# Systematic Review of Mind-Body Interventions to Treat Myalgic Encephalomyelitis/Chronic Fatigue Syndrome

**DOI:** 10.3390/medicina57070652

**Published:** 2021-06-24

**Authors:** Samaneh Khanpour Ardestani, Mohammad Karkhaneh, Eleanor Stein, Salima Punja, Daniela R. Junqueira, Tatiana Kuzmyn, Michelle Pearson, Laurie Smith, Karin Olson, Sunita Vohra

**Affiliations:** 1Department of Pediatrics, Faculty of Medicine & Dentistry, University of Alberta, Edmonton, AB T6G 1C9, Canada; khanpour@ualberta.ca (S.K.A.); punja@ualberta.ca (S.P.); junqueir@ualberta.ca (D.R.J.); 2Institute of Health Economics, Edmonton, AB T6X 0E1, Canada; mk4@ualberta.ca; 3Department of Psychiatry, Faculty of Medicine, University of Calgary, Calgary, AB T2T4L8, Canada; espc@eleanorsteinmd.ca; 4Patient Research Partner, Retired RN, Patient and Community Engagement Research (PaCER) Program Graduate, University of Calgary, Calgary, AB T2P 1B2, Canada; tkuzmyn@telus.net; 5Patient Research Partner, MAPC, CEO Wunjo IS, Calgary, AB T3K 4N8, Canada; michelle@wunjo-is.com; 6Patient Research Partner, Calgary, AB 95060, Canada; lauriem.smith@gmail.com; 7Faculty of Nursing, University of Alberta, Edmonton, AB T6G 1C9, Canada; karin.olson@ualberta.ca; 8Departments of Pediatrics and Psychiatry, Faculty of Medicine & Dentistry, University of Alberta, Edmonton, AB T6G 1C9, Canada

**Keywords:** myalgic encephalomyelitis/chronic fatigue syndrome, mind-body interventions, systematic review, adults

## Abstract

*Background and Objectives*: Myalgic Encephalomyelitis/Chronic Fatigue Syndrome (ME/CFS) is a chronic condition distinguished by disabling fatigue associated with post-exertional malaise, as well as changes to sleep, autonomic functioning, and cognition. Mind-body interventions (MBIs) utilize the ongoing interaction between the mind and body to improve health and wellbeing. Purpose: To systematically review studies using MBIs for the treatment of ME/CFS symptoms. *Materials and Methods*: MEDLINE, EMBASE, CINAHL, PsycINFO, and Cochrane CENTRAL were searched (inception to September 2020). Interventional studies on adults diagnosed with ME/CFS, using one of the MBIs in comparison with any placebo, standard of care treatment or waitlist control, and measuring outcomes relevant to the signs and symptoms of ME/CFS and quality of life were assessed for inclusion. Characteristics and findings of the included studies were summarized using a descriptive approach. *Results*: 12 out of 382 retrieved references were included. Seven studies were randomized controlled trials (RCTs) with one including three reports (1 RCT, 2 single-arms); others were single-arm trials. Interventions included mindfulness-based stress reduction, mindfulness-based cognitive therapy, relaxation, Qigong, cognitive-behavioral stress management, acceptance and commitment therapy and isometric yoga. The outcomes measured most often were fatigue severity, anxiety/depression, and quality of life. Fatigue severity and symptoms of anxiety/depression were improved in nine and eight studies respectively, and three studies found that MBIs improved quality of life. *Conclusions*: Fatigue severity, anxiety/depression and physical and mental functioning were shown to be improved in patients receiving MBIs. However, small sample sizes, heterogeneous diagnostic criteria, and a high risk of bias may challenge this result. Further research using standardized outcomes would help advance the field.

## 1. Introduction

Myalgic encephalomyelitis/chronic fatigue syndrome (ME/CFS) is a chronic condition distinguished by disabling fatigue associated with multiple symptoms including post-exertional malaise, orthostatic intolerance, pain, sleep problems, and impaired cognitive and immune functions [1]. While the true prevalence is unknown, Johnston et al., estimated the pooled prevalence of ME/CFS to be 3.28% and 0.76% according to self-reporting and clinical assessment, respectively [2]. In Canada, 1.4% of people older than 12 years old [3] suffer from ME/CFS. Patients report post-exertional malaise (69–100%), muscle pain (63–95%), impaired memory or concentration (88%), non-restorative sleep (87%), joint pain (55–85%), and sore throat (62%) [1,4]. Health-related quality of life in ME/CFS patients is consistently reported as significantly lower than otherwise healthy populations with regards to physical and mental health, self-care, and ability to perform usual activities [5,6]. Not surprisingly, ME/CFS reduces patients’ abilities to carry out normal working activities leading to higher unemployment rates [7]. It is estimated that annual household and labor force productivity of ME/CFS patients are decreased by 37% and 54%, respectively, costing an approximate annual loss of $9.1 billion in the United States (US) [8]. ME/CFS patients, their families and employers endure a high financial burden estimated to be between $18 to $51 billion annually in the US [9].

Despite extensive research, the etiology and pathophysiology of ME/CFS have not yet been fully understood. Disruptions in the autonomic nervous system, hypothalamic-pituitary-adrenocortical (HPA) axis, and immune system were shown in several studies [10,11]. Metabolic and mitochondrial dysfunction and abnormal gut microbiota were also shown to be interconnected with the above dysregulation [11]. A recent systematic review of neuroimaging studies showed inconsistent but widespread abnormalities in white matter, functional connectivity, and morphological changes of the autonomic nervous system [12].

With no specific etiology, there is no gold standard method to diagnose ME/CFS to date. A recent systematic review of diagnostic methods by Haney et al., identified nine case definitions [13]. Due to the lack of a biomarker, most of the case definitions require other competing diagnoses to be ruled out [14,15]. In the literature, the term myalgic encephalomyelitis (ME) [16] was used earlier than the term chronic fatigue syndrome (CFS) [17]. The Canadian case definition published in 2003 required post-exertional malaise as an essential symptom in these patients and recommended the umbrella term ME/CFS [18], used in this systematic review. 

There is no cure for ME/CFS nor any FDA or Health Canada approved medication to treat it [14,19], therefore the focus tends to be on managing and minimizing the symptoms and improving quality of life. A variety of conventional and complementary therapies have been used to mitigate the symptoms of ME/CFS. As in other chronic conditions, long-term pharmacological interventions may have significant impacts on patients and their families in terms of adverse effects and financial burden [20,21]. Non-pharmacological options are of interest to patients as they may be less expensive and have fewer associated adverse effects. 

Systematic reviews have shown low strength of evidence for the effectiveness of different complementary therapies [19], cognitive-behavioral therapy (CBT), counseling and behavioral therapies [14,22], and graded exercise therapy [23] for improvement of fatigue, physical functioning, sleep, and quality of life in patients with ME/CFS.

Mind-body approaches utilize the interactions between the brain, mind, and body, and behavior to improve health and wellbeing [24]. Using these interconnections strengthens self-awareness and self-care and helps to improve mood, quality of life, and increase one’s ability to cope. Examples of mind-body therapy interventions (MBIs) include progressive muscle relaxation, guided imagery, hypnosis, meditation, mindfulness, Tai chi, yoga, and biofeedback. Newer approaches are using the brain’s ability to change (i.e., neuroplasticity) associated with repeated, purposeful thoughts, feelings or behaviors [25]. The science behind how mind-body therapies work is expanding. It has been shown that the brain and body communicate in multiple directions using neurotransmitters/neuropeptides, hormones, and cytokines and MBIs may be influencing physical health by affecting these interactions [24,26]. 

Considering the complex nature of ME/CFS and the involvement of psycho-neuroendocrine and immune systems, these patients are an ideal population for evaluating MBIs. Furthermore, by enhancing self-knowledge and patients’ abilities to work through their problems and reduce stress, MBIs may improve their quality of life and wellbeing [27].

Several MBIs such as mindfulness-based stress reduction (MBSR), mindfulness-based cognitive therapy (MBCT), yoga, and Qigong have been studied in ME/CFS patients, but to our knowledge, have not yet been included in any systematic review or meta-analysis. There are some promising results to improve anxiety, fatigue, depression, quality of life, and physical functioning [28,29,30,31,32] in ME/CFS. In this systematic review, we evaluated the effectiveness and safety of MBIs that were studied in individuals diagnosed with ME/CFS. The results of this review will inform the design and methodology of future randomized controlled trials. 

### Objectives

The objectives of this study were to systematically review studies of MBIs for the treatment of ME/CFS symptoms and to report any adverse events reported for these approaches in ME/CFS patients.

## 2. Materials and Methods

We followed the Preferred Reporting Items for Systematic Reviews and Meta-Analysis (PRISMA) guidelines [33]. The protocol of this systematic review was registered at PROSPERO (CRD42018085981).

### 2.1. Population, Intervention, Control, Outcome- Study Design (PICO-S)

The population of interest was adults (≥18 years old) diagnosed or symptom-matched with one of the ME/CFS case definitions (Appendix A, Table A1). Patients with any other conditions were included in this review, as long as they were diagnosed with ME/CFS. Interventions of interest included any of the MBIs listed in Table 1 and any placebo, the standard of care treatment or waiting list as a control group. To be eligible for inclusion, multiple-arm interventional studies were also required to have at least one of the control groups mentioned above. 

All outcomes relevant to the signs and symptoms of ME/CFS and quality of life were considered. The outcomes included fatigue, sleep refreshment, pain, anxiety (stress, nervousness, etc.), depression (mood, hopefulness, and helplessness), quality of life, performance (physical, mental, emotional), work-related outcomes (employment, income, etc.), and physical health symptoms such as sore throat, tender lymph nodes, and muscle weakness (Table 1). 

Study designs eligible for inclusion were parallel/cross-over/N-of-1 randomized controlled trials (RCTs), controlled clinical trials (CCTs), single-arm experimental (within subject control group), controlled before and after studies, or cohort studies.

### 2.2. Search Methods

Five electronic databases (MEDLINE, EMBASE, CINAHL, PsycINFO, and Cochrane Register of Controlled Trials (CENTRAL)) were searched from inception to September 2020. Search terms were based on those presented in Table 1; an example is found in Appendix B. No limitation was implemented in terms of publication dates. English language restriction was applied. The reference lists of included studies, and systematic reviews, were reviewed to identify additional studies.

### 2.3. Selection of Studies

Two review authors (MK, DJ) independently screened all the titles and abstracts retrieved from the search in order to identify those that may meet the inclusion criteria. They classified studies as being relevant, possibly relevant and irrelevant. Three reviewers (MK, DJ, SKA) independently assessed the full texts of all relevant and possibly relevant studies to assess inclusion. Discrepancies were resolved by referring to a senior review author (ES, SV). 

### 2.4. Data Collection

Standardized data extraction forms were used to extract data from full-text articles. Extracted data included general characteristics of the study (first author, publication year, country, settings, design), sample size, age and sex distribution in groups, diagnosis methods, type of MBI and other relevant data including frequency and duration, control (active or passive), primary outcome, secondary outcomes, primary and secondary measurement tools, length of study, follow up period, statistically significant outcomes, and adverse events reported. Data extraction was completed by one reviewer (DJ) and independently verified by a second reviewer (SKA). Disagreements between the authors were resolved by discussion until consensus was reached; if consensus could not be reached, a senior reviewer’s opinion was sought. 

### 2.5. Data Analysis

This systematic review was conducted to determine which outcomes and outcome measures were used in the studies of MBIs for the treatment of ME/CFS patients and whether the interventions were effective. General information of the included studies along with the statistically significant and insignificant outcomes were described. We present the findings of studies using different diagnostic criteria (e.g., Oxford criteria, CDC criteria) separately. We also report whether studies assessed adverse events, their absence or presence, and frequencies. A meta-analysis was not performed due to heterogeneous interventions and outcomes used in the included studies. Cochrane risk of bias assessment tool was used by two independent review authors (SKA, SP) to assess sequence generation, allocation concealment, blinding, incomplete outcome data, selective outcome reporting, and other sources of bias [34] in RCTs. Other study designs including single-arm experimental studies were also appraised by two independent reviewers (SKA, SP) for risk of bias using Cochrane Risk of Bias Assessment Tool for Non-Randomized Studies of Intervention (ACROBAT_NRSI) which was recently renamed ROBINS-I [35]. Domains for assessing the risk of bias in these studies include bias due to confounding, selection of participants, measurement of interventions, a departure from the intended intervention, missing data, measurement of outcomes, and selection of the reported result. 

### 2.6. Patient Involvement

Patient engagement in health research can improve the quality, relevance and impact of the research [36,37]. To recruit patient research partners in this study, a “call for patient representative” letter was developed and distributed among patients, caregivers and advocates. Three patient partners were selected based on their educational background, personal experience, and health status to participate in the study team. They did not receive any financial compensation. They participated regularly in teleconference calls and skype meetings. They also provided feedback and participated in team discussions via email. They contributed to the protocol design, development of the literature search strategy, the condition/diagnosis definitions, and outcome selection. 

## 3. Results

Our search results yielded 382 references. After removing duplicates, 270 were screened using title and abstracts, and 47 references were considered relevant for full-text screening. Considering the a priori inclusion criteria and obtaining additional clarifying information from authors of some of the references, twelve studies (17 reports) were ultimately included [10,28,29,30,38,39,40,41,42,43,44,45]. The flow of studies through the screening process of the review is shown in Figure 1. The excluded studies and the reasons for exclusion are shown in Table A2. 

### 3.1. Characteristics of the Included Studies

Table 2 shows the characteristics of all the included studies. 

### 3.2. Design

Six studies were prospective RCTs with at least one eligible control group [28,30,39,40,41,45]. One manuscript presented a brief report of three studies in which one was a prospective RCT and the other two were single-arm experimental studies [29]. Five additional publications were also single-arm experimental studies [10,38,42,43,44].

### 3.3. Population

Participants were all adults diagnosed with ME/CFS (*n* = 564 total; sample size range *n* = 9–150). Six studies used the Center for Disease Control and Prevention (CDC) criteria for the diagnosis of their patients [28,39,40,41,45,51]. One study used the 2003 Canadian criteria [43]. One study used Oxford criteria [29], one study used both CDC and Oxford criteria [30] and three studies used a combination of CDC criteria with 2003 or 2005 versions of Canadian criteria and 2011 international consensus criteria [10,42,44]. 

The healthcare settings included outpatient settings [28,39], community [40,41,43,45], a university hospital clinic [38], department of psychosomatic medicine [10,42] and a specialist ME/CFS unit [30,44]. One study did not report the setting from which their patients were recruited [29].

Three studies were conducted in the United Kingdom [29,30,39], three in Japan [10,28,42], two were conducted in Hong Kong, China [40,41], and one each in Belgium [38], Norway, Sweden, and USA [43,44,45].

### 3.4. Intervention

A variety of different interventions were implemented in the included studies comprising mindfulness-based stress reduction/mindfulness-based cognitive therapy (MBSR/MBCT) [29], MBCT [30,43], relaxation therapy [39], relaxation imagery [38], Qigong exercise training [41], Baduanjin Qigong [40], and isometric yoga [28], seated isometric yoga [42], recumbent isometric yoga [10], acceptance and commitment therapy [44] and cognitive-behavioral stress management [45]. Treatment duration ranged between 5–12 weeks. 

### 3.5. Comparison

Participants assigned to the control group were either placed on the waiting list [28,29,30,40,41] or received standard medical care [39]. They were advised to keep their usual lifestyle activities including seeking general medical care but not to participate in any activities similar to the intervention of interest. 

### 3.6. Outcomes

Many different outcomes and outcome measures were reported in the included studies. Four studies clearly stated their primary and secondary outcomes/objectives [30,39,40,41]. Fatigue severity was measured by seven studies using Chalder fatigue scale [10,28,29,30,40,41,43]. One study (published as two reports), listed Chalder fatigue scale in one of the reports as the administered questionnaire [51]. In the other report, however, they measured fatigue using patient-rated Likert-type scales [39]. Other studies used either profile of mood state (POMS) [42,45] or multidimensional fatigue inventory (MFI-20) [44].

Eight studies measured anxiety and depression using the Hospital Anxiety and Depression Scale (HADS) [29,30,40,41,43,44,51,91]. Six studies measured quality of life or physical and/or mental functioning using different quality of life outcome measures [28,29,30,41,45,51]. Seven studies measured objective outcomes including ventilatory parameters [38], performance testing by computer programs [51], telomerase activity [70], autonomic nervous system functions, blood biomarkers [42,91], adiponectin levels [83], and microRNA changes [10]. Table 2 describes the details of these outcome measures and the other outcomes measured in the included studies. 

### 3.7. Effects of Interventions

Due to heterogeneous interventions and outcome measures used in the included studies, a meta-analysis was not performed. The statistically significant outcomes reported by these studies are presented in Table 3 and Table 4. Table A3 and Table A4 show the statistically insignificant outcomes.

In comparison to the control group, both mental and physical fatigue scores improved significantly in four included studies using MBCT [30], isometric yoga [28], Qigong exercise [41] and Baduanjin Qigong [40]. Two studies showed within-group fatigue improvement in participants receiving an 8-week mindfulness therapy [29] and in participants receiving a 10-week relaxation program [39] (Table 3 and Table 4)

Anxiety and depression were improved in participants receiving Baduanjin Qigong compared to the controls after 16 sessions (9 weeks) of therapy [40]. Depression was improved in participants after 4 months of Qigong exercise [41] and 8 weeks of MBCT [30] compared to the control groups. Surawy et al. [29] also showed improvement of anxiety after 8 weeks of MBSR/MBCT intervention compared to the control group. 

In comparison to the control group, quality of life improved in participants receiving Qigong exercise [41,70,71] and cognitive-behavioral stress management [45].

Table 3 and Table 4 show the details of all the significant outcomes of the included studies according to the diagnosis of ME/CFS (Oxford or CDC criteria). 

### 3.8. Adverse Events

Seven studies assessed adverse events: Four did not identify any adverse events [30,39,41,43]; and three studies recorded adverse events such as deterioration of their symptoms, muscle ache, palpitation, dizziness, knee pain, backache, fatigue, and nervousness [28,40,44]. Five studies did not report if they assessed adverse events [10,29,38,42,45] 

### 3.9. Risk of Bias in the Included Studies

All the included RCT studies were assessed at a high risk of bias in relation to the lack of blinding of participants and personnel. We were not able to assess the risk of bias in many areas as most of the studies were poorly reported (Figure 2). The risk of bias assessment for the single-arm experimental studies using the ROBINS-I assessment tool is shown in Table A5.

## 4. Discussion

This is the first systematic review of studies using MBIs in patients with ME/CFS. The MBIs used in these studies were mindfulness-based stress reduction and mindfulness-based cognitive therapy, relaxation, Qigong, and yoga. 

The etiology and pathogenesis of ME/CFS are still unknown [1]. Researchers have shown changes in some biological markers [100,101,102,103]. Other studies highlight changes in the hypothalamic-pituitary-adrenal (HPA) axis in these patients [104]. 

It was also suggested that ME/CFS may be a neurophysiological disorder in the brain caused by repeated incidental or unnecessary stimuli in the limbic system, which is known as the threat response/protection center. These stimuli can be emotional, psychological, chemical, and/or physiological and they can keep the threat response center on a continuous high alert [105]. Connections between the amygdala and sympathetic, hypothalamic and other limbic brain systems can initiate a series of stimulations and uncontrolled reactions throughout the whole body, which could be considered as the root cause of CFS symptoms [105]. 

With increasing knowledge based on neuroplasticity and the impact of limbic function on somatic symptoms, the potential mechanisms of MBIs might be explained. There is growing interest in using MBIs and many programs are being offered directly to the public to assist with mental and physical health. One of these programs developed specifically for ME/CFS (25) has shown modest success in functional ability in a clinical audit. Because patients are accessing MBI programs, there is an urgent need for evidence as to whether these programs are having an impact on the core symptoms of ME/CFS or mainly address the secondary dissatisfaction that comes with having a chronic, poorly understood disease for which there is no cure. In this review, the MBIs used in the included studies were quite heterogenous. Two studies used relaxation techniques, five studies used movement-based therapies including different forms of yoga and Qigong and the remaining ones used various forms of mindfulness and cognitive-based approaches. Table A6 describes these interventions briefly.

In this systematic review, we found the most commonly measured outcomes were fatigue severity, anxiety and depression, and quality of life or its components (e.g., physical and mental functioning). When compared to the control group, fatigue severity, mental functioning and anxiety/depression mostly improved in patients receiving MBIs. However, poor reporting, small sample sizes, different diagnostic criteria, and a high risk of bias may challenge this result. It is also worth noting that these symptoms are not specific and can be found not only in some individuals with ME/CFS but also in individuals with many other physical and mental health conditions.

According to the 2015 Institute of Medicine report [1], impaired function, post-exertional malaise and unrefreshing sleep are the core symptoms in ME/CFS patients. None of our included studies, however, measured post-exertional malaise. One study measured sleep using a self-reporting scale which improved after 9 weeks of Qigong exercise [40]. Physical or mental functioning and functional performance were mostly measured using self-report scales and only one study measured performance using objective measures [51]. 

In contrast, anxiety and depression and some cognitive constructs were commonly measured in the included studies. While these symptoms are important, they are secondary and not the key features of ME/CFS. Reporting secondary outcomes while omitting measurement of the core symptoms of a disease may lead to inaccurate conclusions about treatment effectiveness. 

Previous studies have used a variety of definitions for the diagnosis of ME/CFS. Lack of consensus and competing definitions act as a barrier for research in this field. Most of the studies in this systematic review used the 1994 CDC criteria for the diagnosis of ME/CFS and two studies used Oxford criteria. 

The Oxford criteria were developed at a consensus meeting [46]. They do not require the presence of any symptom other than disabling fatigue. The presence of other symptoms such as immune, autonomic and mood symptoms differentiate ME/CFS from other common medical and psychiatric conditions including major depression. It has long been suspected that the Oxford criteria may therefore fail to exclude individuals with other fatiguing conditions [14,19].

To address this concern, the Agency for Healthy Research and Quality (AHRQ) in the United States conducted a sensitivity analysis in which the outcomes of treatment studies using the Oxford criteria were compared with studies using other criteria (mostly the 1994 CDC Criteria) [14]. They found that whereas most studies using the Oxford criteria showed some benefits for CBT, studies using the CDC criteria were mixed with no overall benefit. With regards to graded exercise therapy, exclusion of the trials using the Oxford case definition left insufficient evidence about the effectiveness of graded exercise therapy on any outcome. Studies of other therapies were not affected as primary studies had small sample sizes and a high risk of bias. These findings confirm that the choice of inclusion criteria impacts study outcomes. The AHRQ concluded that future research should retire the use of the Oxford case definition. The National Institutes of Health held a consensus workshop to guide the future of ME/CFS research [19]. For similar reasons as the AHRQ, they also recommended that the Oxford Criteria should be retired. 

The 1994 CDC criteria also have significant drawbacks. They require four out of eight criteria but none are mandatory. This means two subjects identified with these criteria may have no symptoms in common with each other—one might have four and the other, another four. Moreover, minor symptoms may overlap with the symptoms of psychiatric disorders including major depression [14]. 

The Institute of Medicine [1] has proposed diagnostic criteria which are very similar to the Canadian Consensus Criteria [88]. They require patients to have moderate, substantial or severe disabling fatigue, post-exertional malaise and unrefreshing sleep for at least half of the time and one of the cognitive impairments or orthostatic intolerance symptoms. Conclusions about the effectiveness of interventions will be possible once studies use the same diagnostic criteria and measure core outcomes using standardized measures. 

### 4.1. Strengths and Limitations

Assessment of a broad range of mind-body approaches and outcomes in a systematic fashion was one of the main strengths of this systematic review. Engaging patients in the process of designing the review protocol and in reviewing the findings increase the applicability and relevance of the findings of this study. 

As we found a diverse range of interventions and outcomes across the included studies; we were not able to perform a meta-analysis. We also may have missed some relevant information by including only studies published in the English language. 

### 4.2. Research Implications


As recommended by the Institute of Medicine report, using objective measures is a priority in studies of ME/CFS. There are several symptoms such as post-exertional malaise, cognitive dysfunction, orthostatic intolerance, and changes including impaired immune function and abnormal brain functions that could be measured objectively.Future RCTs will benefit from larger sample sizes. Investigators must use an appropriate randomization method and ensure outcome assessors are blinded to the group identity of the participants. They should measure and report the outcomes specified in their protocol in order to avoid selective reporting.


## 5. Conclusions

In this systematic review, we described the current literature on MBIs for the treatment of ME/CFS. Future clinical trials will benefit from the findings of this study in terms of what outcomes and outcome measures are mostly used in previous studies. We showed that the included studies did not report measuring post-exertional malaise as a core outcome of ME/CFS. On the other hand, fatigue severity, anxiety/depression and mental functioning were shown to be improved in the patients receiving MBIs. However, poor reporting, small sample sizes, different diagnostic criteria, and a high risk of bias may challenge this result. We highlight the need for further research to use objective and standardized outcomes and outcome measures for making definitive conclusions.

## Figures and Tables

**Figure 1 medicina-57-00652-f001:**
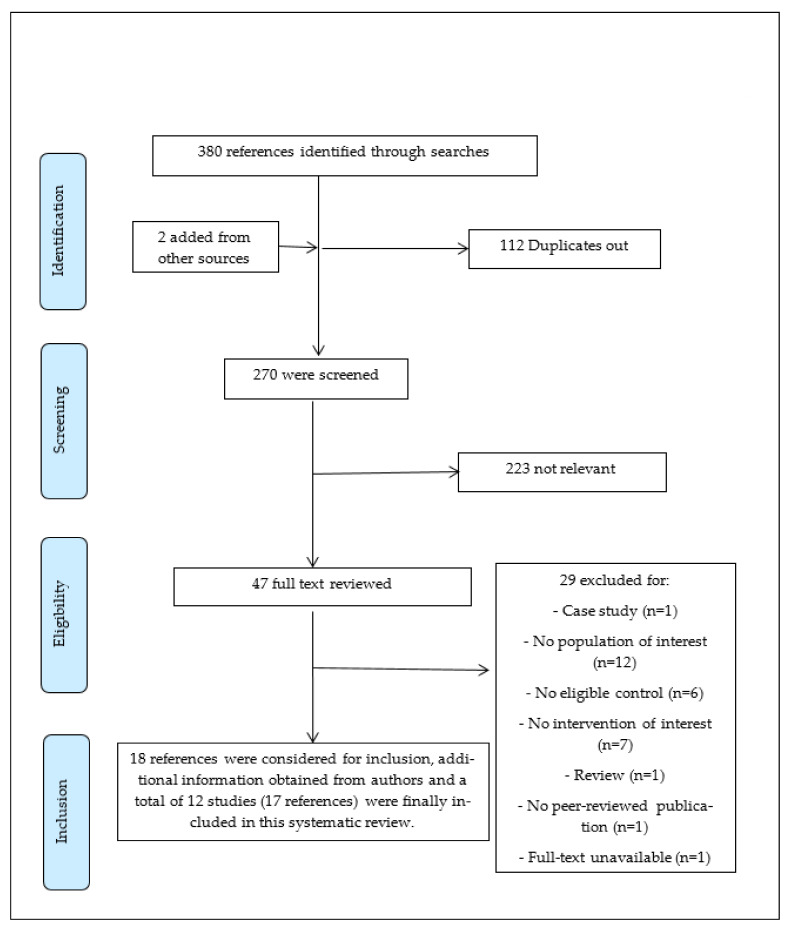
Adapted version of PRISMA flow diagram of study selection for the ME/CFS systematic review.

**Figure 2 medicina-57-00652-f002:**
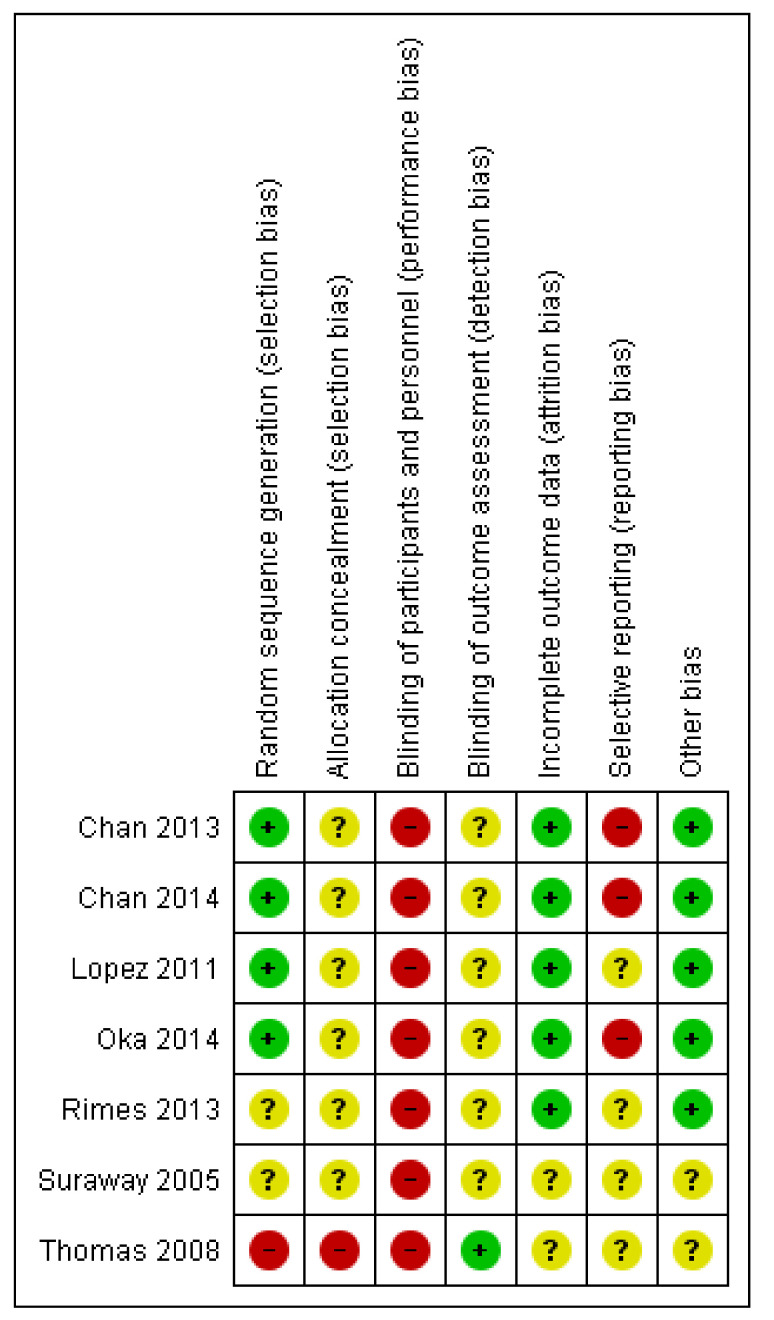
Risk of bias summary: review authors’ judgments about each risk of bias item for each included study.

**Table 1 medicina-57-00652-t001:** Criteria for selecting studies.

Population	Patients with a diagnosis of CFS, ME, and ME/CFS including: Patients who were previously treated Patients who are previously untreated Adults (≥18 years)
Interventions	Mind-body interventions (alone or in combination) including: Art Therapy Autogenic training Biofeedback/neurofeedback Breathing exercise Cognitive restructuring Dynamic Neural Retraining System Emotional Freedom Techniques (EFT) Eye movement desensitization and reprocessing (EMDR) Guided imagery Hypnotherapy/self-hypnosis Meditation (mindfulness, mantra, guided, transcendental) Mindfulness-based cognitive therapy (MBCT) Mindfulness-based Stress Reduction (MBSR) Music therapy Neurolinguistic programming Prayer/spirituality Psychological flexibility Qigong Relaxation therapy (relaxation response, progressive muscle relaxation) Tai Chi Visualization Yoga
Comparators	One or more of the following control conditions including: Placebo Standard of care treatments Waitlist
Outcomes	Any single or combination of, but not limited to, the following outcomes: Fatigue (energy, motivation) Refreshing sleep Pain Anxiety (stress, nervousness, etc.,) Depression (mood, hopefulness, helplessness) Quality of life Performance (physical, mental, emotional) Work-related outcome (employment, income, etc.) Changes in physical health such as sore throat, tender lymph nodes, and muscle weakness
Study Design	Parallel/Cross-over randomized controlled trials (RCTs) Controlled clinical trials (CCTs) Controlled before and after studies Single-arm interventional studies (within subject control group) Cohort
Other	English language

**Table 2 medicina-57-00652-t002:** General characteristics of the included studies.

First Author, Year, Country	Setting	Design, Sample Size (Enrolled/Completed/Analyzed), Treatment Duration	Study Population (Diagnosis, Age, Gender)	Mind-Body Intervention, Frequency, Duration, Self-Practice	Control Group	Outcome, Measurement Tool and Validity
Surawy, Ch., 2005, UK [29]	Not reported	A series of exploratory studies: Study 1 Design: RCT, Sample size: Intervention: 9/9/9, Control: 9/8/8 Treatment duration: 8 weeks Study 2 Design: single-arm trial, Sample size: 12/9/9,Treatment duration: 8 weeks Study 3 Design: single-arm trial, Sample size: 11/9/9, Treatment duration: 8 weeks and a follow-up period of 3 months	Patients diagnosed with Oxford criteria [46] Study 1 Age range: 18–65 y/o, 56% female Study 2 Age range: 18–65 y/o, 75% female Study 3 Age range: 18–65 y/o, 64% female	MBSR/MBCT Frequency: Once a week, Duration: Not reported, Self-practice: Not reported	Study 1: Wait list Study 2: No control group Study 3: No control group	Study 1, 2, and 3 Anxiety and Depression: Hospital Anxiety and Depression Scale (HADS) [47] Fatigue Severity: Chalder’s Fatigue Scale [48] Quality of Life: SF36 physical functioning [49] Study 2 and 3 Effect of fatigue on quality of life: Fatigue impact scale [50]
Thomas, M., 2006 and 2008, UK [39,51]	Outpatient clinics	Design: RCT, Sample size: Intervention (relaxation group): 14/14/14, Control: 9/9/9 Treatment duration: 10 weeks Follow-up: 6 months	Patients diagnosed with CFS by CDC diagnostic criteria for CFS [52] Age (mean ± SD): Intervention (relaxation): 45.7 ± 12.5, Control: 46.2 ± 11.04, Intervention (relaxation): 71.4% female, Control: 66.7% female	Relaxation therapy Frequency: Once a week, Duration: 1 h, Self-practice: Not reported	Standard medical care	Report 1 Illness history: Beck Depression Inventory [53] Centre for Epidemiological Studies-Depression Scale [54], Chalder Fatigue Scale [48], Cognitive Failures Questionnaire [55], Cohen–Hoberman Index of Physical Symptoms [56], Current State of Health [57], Fatigue Problem Rating Scale [58], Hospital Anxiety and Depression Scale [47], MOS SF-36 [49] Perceived Stress [56] Positive and Negative Affect [59], Profile of Fatigue Related Symptoms [60], Sleep Questionnaire [57], Symptom Check List [57] Mood testing: Alertness, hedonic tone and anxiety: measured using 18 computerized visual analogue mood scales Performance testing: Word recall, reaction time, vigilance tasks using a Viglen Dossier laptop computer connected to a simple 3-button response box [57] Report 2 Primary outcome: Functional performance: Karnofsky performance scale [61] Secondary outcome: Global measures of illness and satisfaction with treatment (including improvement and changes in fatigue and disability)
Bogaerts, K., 2007, Belgium [38]	University hospital clinic	Design: Single-arm trial, Sample size: 30/30/30 Treatment duration: Single time imagery trial	Patients diagnosed with CFS by CDC diagnostic criteria for CFS [52]	Relaxation imagery Frequency: onceDuration: less than 5 min Self-practice: NA	No control group	Ventilatory measures: Pet CO_2_ Subjective measures: Degree of fatigue, imagery vividness, concentration ability on the scripts and similarity of evoked feelings with daily life feelings: 9-point rating scale Positive and negative affectivity: Positive and Negative Affect Schedule (PANAS) [62] Hyperventilation complaints: Symptom checklist [63] Chronic fatigue acceptance: Acceptance Chronic Fatigue Test (ACFT) Tendency to worry: Penn-State Worry Questionnaire (PSWQ) [64] Valence, arousal and dominance: Self-assessment Manikin [65]
Lopez, C., 2011. USA [45]	Physician referrals, community	Design: RCT Sample size: 69/58/58 Treatment duration: 12 weeks	Patients diagnosed with CFS by CDC diagnostic criteria for CFS [52], Age (mean ±SD): 45.9 ± 9.3 88.4% female	Cognitive-behavioral stress management Frequency: Weekly Duration: Two hours Self-practice: Workbook and relaxation tapes	Psychoeducation (half-day seminar)	Distress: Perceived Stress Scale (PSS) [66], Profile of Mood States (POMS) [67] Quality of life: Quality of Life Inventory (QOLI) [68] CFS symptoms: CDC Symptom Inventory for Chronic Fatigue Syndrome [69]
Chan, J., 2013, Hong Kong [41] (characteristics of Ho et al., 2012 [70] as the preliminary study and Li et al., 2015 [71] as the study conducted on a subset of participants suffering from bereavement are reported here as well)	Community	Design: RCT, Sample size: Ho, R., 2012 report: Intervention: 35 * /27/33 Control group: 35 ** /25/31 Chan, J., 2013 (main report): Intervention: 77/53/72 ^ Control: 77/58/65 ^^ Li, J., 2015, report: Intervention: 22/22/22 Control: 24/24/24 Treatment duration: 5 consecutive weeks training sessions + 12 weeks home-based qigong exercise (4 months in total) Li et al., however, reported their findings after three months of intervention.	Patients diagnosed with CFS by CDC diagnostic criteria for CFS [52] Age (mean ± SD), % female: Ho, R., 2012 report: Intervention: 42.1 ± 7.3, Control: 42.5 ± 5.5, Intervention: 75.8% female, Control: 83.9% female Chan, J., 2013 (main report): Intervention: 42.4 ± 6.7, Control: 42.5 ± 6.4, Intervention: 72.2% female, Control: 81.5% female Li, J., 2015, report: Patient with CFS had been bereaved within the previous 2 years. Age (median, range): Intervention: 46 (23–52), Control: 45 (32–51), Intervention: 86.4% female, Control: 87.5% female	Qigong exercise training (Wu Xing Ping Heng Gong), Frequency: twice a week, Duration: 2 h, Self-practice: 30 min, every day at home	Waitlist	Chan, J., 2013 (main report): Primary outcome: Fatigue severity: Chalder’s Fatigue Scale [48,72] Secondary outcomes: Anxiety and Depression: Hospital Anxiety and Depression Scale (HADS) [47,73] Ho, R., 2012 report: In addition to fatigue severity, they measured Physical functioning and mental functioning: the Chinese version of the Medical Outcomes Study 12- Item Short-Form Health Survey [74,75] as their primary outcome and Telomerase Activity as their secondary outcome. Li, J., 2015, report: In addition to fatigue severity and anxiety and depression, they measured quality of life: Short form health survey (SF-12) [74,75] and Spiritual well-being: the “spirituality” subscale of the Body-Mind-Spirit Well-being Inventory (BMSWBI-S) [76]
Rimes, K., 2013, U.K. [30]	A specialist National Health Service CFS Unit	Design: RCT, Sample size: Intervention group: 18/16/16 Control group: 19/19/19 Treatment duration: Introductory session + 8 weeks Follow-up: at 2 months, and at 6 months for MBCT group only	Patients diagnosed with CFS by Fukuda et al. [52] criteria or Oxford criteria [46] Age (mean ± SD): Intervention: 41.4 ± 10.9, Control: 45.2 ± 9.4, Intervention:75% female, Control: 89.5% female	MBCT, Frequency: Once a week, Duration: 2.25 h, Self-practice: Home practice with the support of CDs.	Waitlist	Primary outcome: Fatigue: Chalder Fatigue Scale [48] Secondary outcomes: Impairment: The Work and Social Adjustment Scale [77] Physical Functioning: Physical Functioning (PF-10) scale) [78,79] Beliefs about Emotions: Beliefs about Emotions Scale [80] Self-Compassion: Self-Compassion Scale [81] Mindfulness: Five-Facet Mindfulness Questionnaire [82] Anxiety and Depression: Hospital Anxiety and Depression Scale (HADS) [47] All-or-Nothing Behaviour and Catastrophic Thinking about Fatigue: five-item subscale of the Cognitive and Behavior Responses to Symptoms Questionnaire (Moss-Morris and Chalder, in preparation; King’s College London, UK) Acceptability and Engagement: Record of class attendance and amount of home practice undertaken
Chan, J., 2014 and 2017, Hong Kong [40,83]	Community	Design: RCT, Sample size: Report 1: Intervention: 75/57/75 Control: 75/58/75 Report 2: Intervention:46 Control: 62 Treatment duration: 9 consecutive weeks Follow-up: 3-month post-intervention.	Patients diagnosed with CFS by CDC diagnostic criteria for CFS [52] Report 1: Age (mean ± SD): Intervention: 39.1 ± 7.8, Control: 38.9 ± 8.1, Intervention: 61.3% female, Control: 82.7% female Report 2 Age (mean ± SD): 39 ± 7.9 All females	Qigong exercise: Baduanjin Qigong Frequency: 16 sessions, Duration: 1.5 h, Self-practice: 30 min, every day	Waitlist	Report 1 Primary outcomes: Sleep Quality: Pittsburgh Sleep Quality Index (PSQI) [84,85,86] Fatigue severity: Chalder Fatigue Scale (ChFS) [48,72] Anxiety and Depression: Hospital Anxiety and Depression Scale (HADS) [47,73] Secondary outcome: Dose-response relationship between Qigong exercise and improvements. Global Assessment, Satisfaction Report 2 Anxiety and Depression: Hospital Anxiety and Depression Scale (HADS) [47,73] Plasma Adiponectin Levels
Oka, T., 2014, Japan [28]	Outpatients with CFS who visited the Department of Psychosomatic Medicine of KyushuUniversity Hospital	Design: RCT, Sample size: Intervention: 15/15/15 Control: 15/15/15 Treatment duration: Two months	Patients diagnosed with CFS by CDC diagnostic criteria for CFS [52] Age (mean ±SD): Intervention:38.0 ±11.1, Control:39.1 ± 14.2, Intervention: 80% female, Control: 80% female	Isometric yoga Frequency: every two to three weeks, at least 4 times during the intervention period, Duration: 20 min, Self-practice: With the aid of a digital videodisk and booklet	Waitlist	Acute effects of isometric yoga on fatigue: The fatigue and vigor score of the Profile of Mood States (POMS) questionnaire [67] immediately after the final 20-min yoga session Chronic effects of isometric yoga on fatigue: Chalder’s Fatigue Scale [48] Quality of Life: Medical Outcomes Study Short Form 8, standard version (SF-8) [87]
Sollie, K., 2017. Norway [43]	Community	Design: Single-arm trial Sample size: 10 Treatment duration: Eight weeks with three months follow-up	Patients diagnosed with CFS by Canada criteria [88] Age (mean ± SD):43.5 ± 9.9, 80% female	MBCT, Frequency: Weekly Duration: Two hours Self-practice: Homework with the aid of workbook and CD	No control group	Fatigue: Chalder Fatigue Scale [48] Symptom burden: Likert scale Anxiety and depression: Hospital Anxiety and Depression Scale (HADS) [47] Tendency to ruminate: Ruminative Response Scale [89] Dispositional mindfulness: Five Facet Mindfulness questionnaire [82] Quality of life: Satisfaction with Life Scale (SWLS) [90]
Oka, T., 2018 and 2019, Japan [42,91]	Outpatients with CFS who visited the Department of Psychosomatic Medicine of KyushuUniversity Hospital	Design: Single-arm trial Sample size: 15 Treatment duration: Eight weeks	Patients diagnosed with CFS by CDC diagnostic criteria for CFS [52], the 2011 international consensus criteria for myalgic encephalomyelitis [92] and the 2015 diagnostic criteria for systemic exertion intolerance disease [1] Age (mean ± SD): 38.0 ± 11.1 80% female	Sitting isometric yoga Frequency: Biweekly with a yoga instructor Duration: 20 min Self-practice: Daily in-home session	No control group	Report 1: Fatigue and vigor: The fatigue and vigor score of the Profile of Mood States (POMS) questionnaire [67] Autonomic nervous system (ANS) functions: Heart rate and Heart rate variability (HRV) Blood biomarkers: Serum cortisol, DHEA-S, TNF-α, IL-6, IFN-α, IFN-γ, PRL, total carnitine, free carnitine, and acylcarnitine, and plasma TGF-β1, BDNF, MHPG, and HVA Report 2: Fatigue severity: Chalder fatigue scale (FS) score [48] Levels of the blood biomarkers: Cortisol, DHEA-S, TNF-α, IL-6, prolactin, carnitine, TGF-β1, BDNF, MHPG, HVA, and α-MSH The autonomic nervous functions: Heart rate (HR) and HR variability Alexithymia: The 20-item Toronto Alexithymia Scale (TAS-20) [93] Anxiety and depression: Japanese version of the Hospital Anxiety and Depression Scale (HADS) [47]
Jonsjo, M., 2019, Sweden [44]	Tertiary specialist clinic	Design: Single-arm trial Sample size: 40/32/32 Treatment duration: 13 sessions with three- and six-month follow-up	Patients diagnosed with CFS according to CDC [52] and 2003 Canadian criteria for ME/CFS [88] Age (mean ± SD): 49.02 ± 10.78 76.7% female	Acceptance and commitment therapy Frequency: Weekly to biweekly depending on illness severity (13 sessions) Duration: 45 min Self-practice: Home assignments	No control group	Primary outcomes: Disability: The pain disability index [94] Psychological inflexibility: The Psychological Inflexibility in Fatigue Scale (PIFS) [95] Secondary outcomes: ME/CFS symptoms and severity: 5-point scale Fatigue: The Multidimensional Fatigue Inventory (MFI-20) [96] Anxiety and depression: The Hospital Anxiety and Depression Scale (HADS) [47] Dimensions of mental and physical health: SF-36 Health Survey [79] Health-related quality of life: EQ-5D-3L [97]
Takakura, S., 2019, Japan [10]	Outpatients with CFS who visited the Department of Psychosomatic Medicine of Kyushu University Hospital	Design: Single-arm trial Sample size: 9 Treatment duration: Three months	Patients diagnosed with CFS according to the 1994 Fukuda case definition of CFS [52], the 2011 International Consensus Criteria for ME [92], and the 2015 diagnostic criteria for systemic exertion intolerance disease [1] Age (mean ± SD): 37.2 ± 9.9 All female	Recumbent isometric yoga Frequency: Every two to four weeks Duration: 20–30 min depending on patient’s preference Self-practice: In-home daily sessions	No control group	Fatigue: Japanese version of 11 item Chalder Fatigue Scale [98,99] Human microRNA

* 2 dropped out before the intervention ** 4 dropped out before the intervention ^ 5 dropped out before the intervention ^^ 12 dropped out before the intervention.

**Table 3 medicina-57-00652-t003:** Significant outcomes in the included studies using CDC, Canadian and international consensus criteria for diagnosing CFS.

Intervention Type	First Author, Year		Outcome (Assessed by)	Comparison Groups	Comparison Time Point	*p*-Value
Relaxation-based	Thomas, M., 2006 and 2008 [39,51]	Report 1 (2006)	Alertness (as part of a subjective mood scale)	Relaxation group (pre-post)	Post follow-up (6 months)	<0.027
			Anxiety (as part of a subjective mood scale)	Relaxation group (pre-post)	Post follow-up (6 months)	<0.002
			Current state of health (self-reporting scale)	Relaxation group (pre-post)	Post-treatment (10 weeks)	Reported significant (value not reported)
		Report 2 (2008)	Performance score-10% improvement (Karnofsky scale)	Relaxation group (pre-post)	Post-treatment (10 weeks)	Reported significant (value not reported)
			Global measures of health: overall condition (Likert-type scale)	Relaxation group (pre-post)	Post-treatment (10 weeks)	Reported significant (value not reported)
			Global measures of health: Fatigue levels (Likert-type scale)	Relaxation group (pre-post)	Post-treatment (10 weeks)	Reported significant (value not reported)
			Global measures of health: Fatigue levels (Likert-type scale)	Relaxation group (pre-post)	Post follow-up (6 months)	Reported significant (value not reported)
			Global measures of health: reduction in disability (Likert-type scale)	Relaxation group (pre-post)	Post-treatment (10 weeks)	Reported significant (value not reported)
			Global measures of health: reduction in disability (Likert-type scale)	Relaxation group (pre-post)	Post follow-up (6 months)	Reported significant (value not reported)
	Bogaerts K., 2007 [38]		PetCO_2_	Relaxation imagery (pre-post)	Post intervention	<0.01
Cognitive-based	Lopez C., 2011 [45] (Cognitive restructuring)		Distress (Perceived stress scale)	CBSM compared to control	Time X group **	0.03
			Total mood disturbance (Profile of Mood States (POMS)	CBSM compared to control	Time X group **	0.05
			Quality of life (QOLI Category)	CBSM compared to control	Time X group **	0.002
			Quality of life (QOLI Raw score)	CBSM compared to control	Time X group **	0.05
			Quality of life (QOLI Total score)	CBSM compared to control	Time X group **	0.05
			CFS symptoms (Total CDC symptom severity)	CBSM compared to control	Time X group **	0.04
	Sollie K., 2017 [43] (Mindfulness-based cognitive therapy)		Fatigue (Chalder Fatigue Scale)	MBCT (pre-post)	Post intervention (8 weeks)	p value not reported, medium effect size was reported (d = 0.56)
			Anxiety (HADS)	MBCT (pre-post)	Post intervention (8 weeks)	p value not reported, medium to large effect size was reported (d = 0.68)
			Anxiety (HADS)	MBCT (pre-post)	Post follow-up (3 months)	p value not reported, medium effect size was reported (d = 0.48)
			Dispositional mindfulness (Five Facet Mindfulness questionnaire)	MBCT (pre-post)	Post follow-up (3 months)	p value not reported, large effect size was reported (d = 0.77)
	Jonsjo, M., 2019 [44] (Psychological flexibility)		Disability (Pain disability index)	ACT (pre-post)	Post intervention (after 13 sessions)	0.000
			Psychological flexibility (Psychological inflexibility fatigue scale)	ACT (pre-post)	Post intervention (after 13 sessions)	0.000
			CFS symptoms	ACT (pre-post)	Post intervention (after 13 sessions)	0.017
			Anxiety (HADS)	ACT (pre-post)	Post intervention (after 13 sessions)	0.001
			General fatigue (MFI-20)	ACT (pre-post)	Post intervention (after 13 sessions)	0.024
			General fatigue (MFI-20)	ACT (pre-post)	Post intervention to post follow-up (3 months)	0.049
			Physical fatigue (MFI-20)	ACT (pre-post)	Post intervention (after 13 sessions)	0.046
			Mental fatigue (MFI-20)	ACT (pre-post)	Post intervention (after 13 sessions)	0.004
			Reduced activity (MFI-20)	ACT (pre-post)	Post intervention (after 13 sessions)	0.041
			Reduced motivation (MFI-20)	ACT (pre-post)	Post intervention (after 13 sessions)	0.043
			SF-36 physical	ACT (pre-post)	Post intervention (after 13 sessions)	0.009
Movement-based	Chan J., 2013 [41]	Ho, R., 2012 (Preliminary report)	Quality of life: Mental functioning score (MOS SF-12)	Qigong (pre-post)	Post-training (5 weeks)	<0.01
			Quality of life: Mental functioning score (MOS SF-12)	Qigong (pre-post)	Post intervention (4 months)	<0.01
			Quality of life: Mental functioning score (MOS SF-12)	Qigong compared to control	Time X group **	0.001
			Telomerase activity * (Telomerase PCR ELISA)	Qigong (pre-post)	Post intervention (4 months)	<0.05
			Telomerase activity * (Telomerase PCR ELISA)	Qigong compared to control	Time X group **	0.029
		Chan J., 2013 (main report)	Total fatigue score (ChFS)	Qigong (pre-post)	Post intervention (4 months)	<0.001
			Total fatigue score (ChFS)	Qigong compared to control	Time X group **	0.000
			Physical fatigue score (ChFS)	Qigong (pre-post)	Post intervention (4 months)	<0.001
			Physical fatigue score (ChFS)	Qigong compared to control	Time X group **	0.000
			Mental fatigue score (ChFS)	Qigong (pre-post)	Post intervention (4 months)	<0.001
			Mental fatigue score (ChFS)	Qigong compared to control	Time X group **	0.050
			Anxiety score (HADS)	Qigong (pre-post)	Post intervention (4 months)	<0.001
			Depression score (HADS)	Qigong (pre-post)	Post intervention (4 months)	<0.001
			Depression score (HADS)	Qigong compared to control	Time X group **	0.002
		Li J., 2015 (Subset study report)	Spirituality (the spirituality subscale of BMSWBI-S)	Qigong compared to control	Post intervention (3 months)	0.013
			Quality of life: mental component summary (MOS SF-12)	Qigong compared to control	Post intervention (3 months)	0.002
			Quality of life: mental component summary (MOS-SF 12)	Qigong compared to control	Post intervention (change score from baseline to 3 months)	0.002
	Chan, J. 2014 [40] and Chan, J. 2017 [83]	Report 1 (2014)	Sleep quality: total score (PSQI)	Baduanjin Qigong compared to waitlist	Post intervention (change score from baseline to 9 weeks)	<0.05
			Subjective sleep quality (PSQI)	Baduanjin Qigong compared to waitlist	Post intervention (change score from baseline to 9 weeks)	<0.01
			Subjective sleep quality (PSQI)	Baduanjin Qigong compared to waitlist	Post follow-up (change score from baseline to 3 months)	<0.01
			Subjective sleep quality (PSQI)	Baduanjin Qigong compared to wait list	Time X group **	0.002
			Sleep latency (PSQI)	Baduanjin Qigong compared to waitlist	Post intervention (change score from baseline to 9 weeks)	<0.05
			Sleep latency (PSQI)	Baduanjin Qigong compared to waitlist	Time X group **	0.044
			Sleep duration (PSQI)	Baduanjin Qigong compared to waitlist	Post intervention (change score from baseline to 9 weeks)	<0.05
			Total fatigue score (ChFS)	Baduanjin Qigong compared to waitlist	Post intervention (change score from baseline to 9 weeks)	<0.001
			Total fatigue score (ChFS)	Baduanjin Qigong compared to waitlist	Post follow-up (change score from baseline to 3 months)	<0.001
			Total fatigue score (ChFS)	Baduanjin Qigong compared to waitlist	Time X group **	<0.001
			Physical fatigue score (ChFS)	Baduanjin Qigong compared to waitlist	Post intervention (change score from baseline to 9 weeks)	<0.001
			Physical fatigue score (ChFS)	Baduanjin Qigong compared to waitlist	Post follow-up (change score from baseline to 3 months)	<0.001
			Physical fatigue score (ChFS)	Baduanjin Qigong compared to waitlist	Time X group **	<0.001
			Mental fatigue score (ChFS)	Baduanjin Qigong compared to waitlist	Post intervention (change score from baseline to 9 weeks)	<0.001
			Mental fatigue score (ChFS)	Baduanjin Qigong compared to waitlist	Post follow-up (change score from baseline to 3 months)	<0.01
			Mental fatigue score (ChFS)	Baduanjin Qigong compared to waitlist	Time X group **	<0.001
			Anxiety (HADS)	Baduanjin Qigong compared to waitlist	Post intervention (change score from baseline to 9 weeks)	<0.01
			Anxiety (HADS)	Baduanjin Qigong compared to waitlist	Post follow-up (change score from baseline to 3 months)	<0.05
			Anxiety (HADS)	Baduanjin Qigong compared to waitlist	Time X group **	0.016
			Depression (HADS)	Baduanjin Qigong compared to waitlist	Post intervention (change score from baseline to 9 weeks)	<0.001
			Depression (HADS)	Baduanjin Qigong compared to waitlist	Time X group **	<0.001
		Report 2 (2017)	Increase in adiponectin levels	Baduanjin Qigong compared to waitlist	Post intervention (change score from baseline to 9 weeks)	<0.05
			Depression (HADS)	Baduanjin Qigong compared to waitlist	Post intervention (change score from baseline to 9 weeks)	<0.001
			Anxiety (HADS)	Baduanjin Qigong compared to waitlist	Post intervention (9 weeks)	<0.05
	Oka T, 2014 [28]		Fatigue score -acute effect (POMS)	Isometric yoga (pre-post)	Before to after the final 20-min session	<0.001
			Vigor score- acute effect (POMS)	Isometric yoga (pre-post)	Before to after the final 20-min session	<0.01
			Physical fatigue score (ChFS)	Isometric yoga (pre-post)	Post intervention (2 months)	0.004
			Physical fatigue score (ChFS)	Isometric yoga compared to control	Time X group **	0.009
			Mental fatigue score (ChFS)	Isometric yoga (pre-post)	Post intervention (2 months)	0.004
			Mental fatigue score (ChFS)	Isometric yoga compared to control	Time X group **	0.007
			Total fatigue score (ChFS)	Isometric yoga (pre-post)	Post intervention (2 months)	0.002
			Total fatigue score (ChFS)	Isometric yoga compared to control	Time X group **	0.003
			Quality of life: bodily pain (SF-8)	Isometric yoga (pre-post)	Post intervention (2 months)	0.0001
			Quality of life: general health perception (SF-8)	Isometric yoga (pre-post)	Post intervention (2 months)	0.0021
			Quality of life: Physical component summary (SF-8)	Isometric yoga (pre-post)	Post intervention (2 months)	0.024
	Oka, T., 2018 and Oka, T., 2019 [42,91]	Report 1 (2018)	Fatigue (POMS)	Acute effects of sitting isometric yoga (pre-post)	Before to after the final 20-min session	0.001
			Vigor (POMS)	Acute effects of sitting isometric yoga (pre-post)	Before to after the final 20-min session	0.002
			Decreased heart rate	Acute effects of sitting isometric yoga (pre-post)	Before to after the final 20-min session	0.047
			Increased high-frequency power of HR variability	Acute effects of sitting isometric yoga (pre-post)	Before to after the final 20-min session	0.028
			Increased serum levels of DHEA-S	Acute effects of sitting isometric yoga (pre-post)	Before to after the final 20-min session	0.012
			Decreased levels of cortisol	Acute effects of sitting isometric yoga (pre-post)	Before to after the final 20-min session	0.016
			Decreased level of TNF-α	Acute effects of sitting isometric yoga (pre-post)	Before to after the final 20-min session	0.035
		Report 2 (2019)	Fatigue (POMS)	Longitudinal effects of sitting isometric yoga (pre-post)	Post intervention (2 months)	0.002
			Depression (HADS)	Longitudinal effects of sitting isometric yoga (pre-post)	Post intervention (2 months)	0.02
	Takakura, S., 2019 [10]		Fatigue (11 score Chalder’s fatigue scale)	Recumbent isometric yoga (pre-post)	Post intervention (3 months)	<0.0001
			Changes in miRNA expression	Recumbent isometric yoga (pre-post)	Post intervention (3 months)	<0.05 (Four miRNAs significantly upregulated and 42 were significantly downregulated)

BMSWBI-S: Body-Mind-Spirit Well-being Inventory, ChFS: Chalder’s Fatigue Scale, HADS: Hospital Anxiety and Depression Scale, MOS SF-12: Medical Outcomes Study 12- Item Short-Form Health Survey, POMS: Profile of Mood States, PSQI: Pittsburgh Sleep Quality Index, SF-8: Medical Outcomes Study Short Form 8, QOLI: Quality of life inventory, MFI-20: Multidimensional fatigue inventory-20, ACT: Acceptance and commitment therapy, MBCT: Mindfulness-based cognitive therapy, CBSM: Cognitive-based stress management. * measure of stress-related damage at a cellular level. ** to test the interaction effect of time and group.

**Table 4 medicina-57-00652-t004:** Significant outcomes in the included studies using Oxford criteria for diagnosing CFS.

Intervention Type	First Author, Year		Outcome (Assessed by)	Comparison Groups	Comparison Time Point	*p*-Value
Mindfulness and cognitive-based	Surawy, Ch., 2005 [29]	Study 1 (RCT)	Anxiety (HADS)	MBSR/MBCT compared to controls	Post treatment (8 weeks)	0.010
		Study 2 (single-arm experimental study)	Anxiety (HADS)	MBCT/MBSR (pre-post)	Post treatment (8 weeks)	0.000
			Fatigue impact: total score (FIS)	MBCT/MBSR (pre-post)	Post treatment (8 weeks)	0.010
		Study 3 (single-arm experimental study)	Anxiety (HADS)	MBCT/MBSR (pre-post)	Post treatment (8 weeks)	0.010
			Anxiety (HADS)	MBCT/MBSR (pre-post)	Post follow-up (3 months)	0.010
			Depression (HADS)	MBCT/MBSR (pre-post)	Post-treatment (8 weeks)	0.010
			Depression (HADS)	MBCT/MBSR (pre-post)	Post follow-up (3 months)	0.050
			Fatigue score (ChFS)	MBCT/MBSR (pre-post)	Post treatment (8 weeks)	0.010
			Fatigue score (ChFS)	MBCT/MBSR (pre-post)	Post follow-up (3 months)	0.000
			Quality of life: physical functioning (SF-36)	MBCT/MBSR (pre-post)	Post treatment (8 weeks)	0.010
			Quality of life: physical functioning (SF-36)	MBCT/MBSR (pre-post)	Post follow-up (3 months)	0.000
			Fatigue impact: total score (FIS)	MBCT/MBSR (pre-post)	Post treatment (8 weeks)	0.020
			Fatigue impact: total score (FIS)	MBCT/MBSR (pre-post)	Post follow-up (3 months)	0.050
	Rimes, K., 2013 [30]		Fatigue (ChFS)	MBCT compared to waitlist	Post treatment (8 weeks)	0.014
			Fatigue (ChFS)	MBCT compared to waitlist	Post follow-up (2 months)	0.033
			Fatigue (ChFS)	MBCT (pre-post)	Post follow-up (6 months)	0.010
			Impairment (The work and social adjustment scale)	MBCT compared to wait list	Post-treatment (8 weeks)	0.04
			Impairment (The work and social adjustment scale)	MBCT compared to waitlist	Post follow-up (2 months)	0.054
			Impairment (The work and social adjustment scale)	MBCT (pre-post)	Post follow-up (6 months)	0.004
			Impairment (The work and social adjustment scale)	MBCT (pre-post)	Between 2- and 6-month follow-up	0.004
			Beliefs about Emotions (Self-reporting scale)	MBCT compared to waitlist	Post treatment (8 weeks)	0.01
			Beliefs about Emotions (Self-reporting scale)	MBCT compared to waitlist	Post follow-up (2 months)	0.012
			Beliefs about Emotions (Self-reporting scale)	MBCT (pre-post)	Post follow-up (6 months)	0.004
			Self-Compassion (Self-reporting scale)	MBCT compared to waitlist	Post treatment (8 weeks)	0.007
			Self-Compassion (Self-reporting scale)	MBCT compared to waitlist	Post follow-up (2 months)	0.006
			Self-Compassion (Self-reporting scale)	MBCT (pre-post)	Post follow-up (6 months)	0.003
			Mindfulness (5 facet mindfulness questionnaire)	MBCT compared to waitlist	Post follow-up (2 months)	0.035
			Mindfulness (5 facet mindfulness questionnaire)	MBCT (pre-post)	Post follow-up (6 months)	0.006
			Mindfulness (5 facet mindfulness questionnaire)	MBCT (pre-post)	Between 2- and 6-month follow-up	0.017
			Catastrophizing (Self-reporting scale)	MBCT compared to waitlist	Post treatment (8 weeks)	0.004
			Catastrophizing (Self-reporting scale)	MBCT (pre-post)	Post follow-up (6 months)	0.012
			All-or-nothing behavior (Self-reporting scale)	MBCT compared to waitlist	Post treatment (8 weeks)	0.005
			All-or-nothing behavior (Self-reporting scale)	MBCT (pre-post)	Post follow-up (6 months)	0.017
			Depression (HADS)	MBCT compared to waitlist	Post-treatment (8 weeks)	0.038

ChFS: Chalder’s Fatigue Scale, HADS: Hospital Anxiety and Depression Scale, MBSR: Mindfulness-based stress reduction, MBCT: Mindfulness-based cognitive therapy, SF-36: 36- Item Short-Form Health Survey.

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
