# Peer review of "Systematic Review of Mind-Body Interventions to Treat Myalgic Encephalomyelitis/Chronic Fatigue Syndrome"

_medicina, 2021, doi:10.3390/medicina57070652_

Round 1

Reviewer 1 Report

The authors did an excellent job addressing all revisions. This will be an excellent contribution to the literature and future research in this area. 

Author Response

We would like to thank the reviewer for the thorough review of our manuscript. 

Reviewer 2 Report

First, I would like to commend the authors' efforts through the revision. However, there are still some issues that have not been resolved.

The authors responded to me that they reported only significant outcomes. But that wasn't listed in their pre-registered protocol (CRD42018085981). Both statistically significant and non-significant results need to be provided to the reader, and each has its own meaning. If all statistically insignificant results are excluded, it can be perceived as a biased report.

According to the above, <3.7. Effects of interventions> can be supplemented further. In other words, it may include results that were not statistically significant.

Table still needs more organization. Please consider presenting the table horizontally.

As the authors point out, several mind-body modalities in this review have great heterogeneity. However, it is questionable whether the authors have had enough discussions considering this heterogeneity.

Author Response

We would like to thank the reviewer for their valuable comments and suggestions which helped us to improve the quality of our manuscript. Attached is our point-by-point response to reviewer’s comments.

This manuscript is a resubmission of an earlier submission. The following is a list of the peer review reports and author responses from that submission.

Round 1

Reviewer 1 Report

First, I am honored to have the opportunity to review this interesting manuscript. I think this study is a well-written, systematic review with a thorough report. However, I think the authors will be able to make it better if they reflect the following items. 1. The search date for this study is October 2017. This is too old to be published in the second half of 2020. Therefore, it is recommended to search for updates. 2. Although I know that the etiology is not clearly identified, it would be nice if there was a description of the potential causes of ME/CFS in the Introduction part. In particular, if the cause and the therapeutic mechanism of MBIs overlap, I think this paper can gain a reinforced justification for conducting a systematic review of MBIs. For example, the authors might mention brain plasticity. 3. As mentioned above, it would be better if more emphasis on why it is necessary to synthesize the evidence for MBIs, why we should be interested in this intervention, and what strengths are there. 4. OBJECTIVES explains “To review all adverse events ~ patients.” However, this study is not intended to look only at the safety profile of MBIs. 5. It would be good to classify the outcomes into primary and secondary according to the importance or clinical relevance. 6. Did the authors not consider the patient's comorbidity in the PICO? 7. Was ROBINS-I also implemented by two independent authors? 8. There was an error in the uploaded PRISMA flow chart. I cannot find some numbers of studies. 9. I think the Tables are too long and contain detailed explanations. It is recommended to further summarize the content or use more abbreviation. 10. The authors may be aware that the treatments investigated are heterogeneous. MBSR/MBCT, relaxation, imagery, qigong, etc. can all belong to MBIs, but each has its own characteristics. For example, MBCT has an element of cognitive therapy, and qigong has an element of physical movement. So, from the reader's point of view, I wonder if each MBI affects different aspects of ME/CFS. In the current manuscript, I think there is a lack of consideration on the effect of the intervention type. I understand that there was not enough data to explain this. However, the authors may be able to provide preliminary data. If even that is not possible, the authors can describe some additional suggestions for future research. 11. The authors stated that statistically significant outcome were presented in 3.7. If so, did the authors not present statistically insignificant outcome? 12. As the authors suggest, studies investigating brain function using neuroimaging for MBIs and ME/CFS may be promising. I expect the authors to describe these available neuroimaging methods in more detail, not simply “abnormal brain functions”.

Reviewer 2 Report

The authors have done excellent work in this systematic review. The validity, reliability and applicability of MBI's in individuals living with CFS is reported in alignment with the findings of the review and is consistent with systematic reviews of MBI's in other populations. This is the first systematic review in these patients and will contribute to the literature and to improving the evidence base for MBI's so they can be integrated appropriately into clinical care and self-management. Thank you for this work.